# Complement factor H protects tumor cell-derived exosomes from complement-dependent lysis and phagocytosis

Ryan T. Bushey[1], Elizabeth B. Gottlin[1], Michael J. Campa[1], Edward F. Patz, Jr.[1,2,3]*

1 Department of Radiology, Duke University Medical Center, Durham, North Carolina, United States of America, 2 Department of Pharmacology and Cancer Biology, Duke University Medical Center, Durham, North Carolina, United States of America, 3 Duke Cancer Institute, Duke University Medical Center, Durham, North Carolina, United States of America

* patz0002@mc.duke.edu

**Data Availability Statement:** All relevant data are within the paper and its Supporting Information files.

## Abstract

Exosomes are a class of extracellular vesicles (EVs) that are mediators of normal intercellular communication, but exosomes are also used by tumor cells to promote oncogenesis and metastasis. Complement factor H (CFH) protects host cells from attack and destruction by the alternative pathway of complement-dependent cytotoxicity (CDC). Here we show that CFH can protect exosomes from complement-mediated lysis and phagocytosis. CFH was found to be associated with EVs from a variety of tumor cell lines as well as EVs isolated from the plasma of patients with metastatic non-small cell lung cancer. Higher levels of CFH-containing EVs correlated with higher metastatic potential of cell lines. GT103, a previously described antibody to CFH that preferentially causes CDC of tumor cells, was used to probe the susceptibility of tumor cell-derived exosomes to destruction. Exosomes were purified from EVs using CD63 beads. Incubation of GT103 with tumor cell-derived exosomes triggered exosome lysis primarily by the classical complement pathway as well as antibody-dependent exosome phagocytosis by macrophages. These results imply that GT103-mediated exosome destruction can be triggered by antibody Fc-C1q interaction (in the case of lysis), and antibody-Fc receptor interactions (in the case of phagocytosis). Thus, this work demonstrates CFH is expressed on tumor cell derived exosomes, can protect them from complement lysis and phagocytosis, and that an anti-CFH antibody can be used to target tumor-derived exosomes for exosome destruction via innate immune mechanisms. These findings suggest that a therapeutic CFH antibody has the potential to inhibit tumor progression and reduce metastasis promoted by exosomes.

## Introduction

Extracellular vesicles (EVs) are mediators of intercellular communication, transporting proteins and nucleic acids from cells of origin to recipient cells and altering their phenotypes [1, 2]. Exosomes, ~30–150 nm diameter EVs of endocytic origin, are of particular interest in cancer as they contribute to oncogenesis by transferring their cargo into other tumor cells, leading

**Funding:** The authors received no specific funding for this work.

**Competing interests:** Drs. Gottlin, Campa, and Patz are founders of Grid Therapeutics, LLC, which is supplying GT103 for a Phase 1b clinical trial. This does not alter our adherence to PLOS ONE policies on sharing data and materials.

to altered gene expression and increased growth, motility, and invasion [3–5]. Exosomes also mediate communication between tumor cells and non-tumor cells, promoting vascularization and formation of premetastatic niches, and are immunosuppressive [6–11]. Exosomal PD-L1 is a powerful contributor to the immunosuppression of T cells and higher expression of PD-L1 on exosomes is associated with poorer outcomes and less responsiveness to checkpoint inhibitors [10, 11].

Complement factor H has been reported to be a component of the proteome of EVs isolated from the plasma of lung adenocarcinoma patients [12] and from highly metastatic hepatocellular carcinoma (HCC) cell lines [13]. It was recently shown that CFH-enriched HCC-derived exosomes promote HCC cell growth, migration and liver metastasis formation and these effects are negated by a tumor-specific antibody to CFH [13, 14]. As CFH protects cells from complement-mediated cytotoxicity (CDC), we hypothesize that exosomes are likewise protected from lysis. Inactivation of CFH and reducing the overall exosome level via complement-mediated destruction might therefore be a viable strategy to inhibit metastasis and reduce the immunosuppressive effects of PD-L1.

The complement system, that is operationally classified into classical, alternative, and lectin pathways, is an important branch of innate immunity [15]. The classical and lectin pathways are initiated by antibodies and pattern recognition proteins, respectively, whereas the alternative pathway is spontaneously active. The key event in all pathways is the formation of a "convertase" enzyme that deposits C3b on the target cell surface, leading to the complement cascade that ultimately releases complement split products and forms a lytic membrane attack complex (MAC). In order to prevent normal host cell damage, the alternative pathway convertase must be regulated. This function is performed by CFH, which promotes the dissociation of the convertase protein complex and prevents the formation of additional convertases. Tumor cells also use CFH to evade destruction by complement [16, 17] and a high level of CFH on tumors is associated with poor survival in lung adenocarcinoma [18].

In this report, we used GT103, a unique, fully human derived monoclonal antibody with apparent specificity for tumor cell-associated CFH, to investigate whether tumor cell-derived exosomes can be destroyed by lysis. In addition, since target cells opsonized by complement or immune complexes are also susceptible to phagocytosis via complement receptor or Fc receptor binding, respectively, we investigated whether these exosomes can be phagocytosed. The GT103 antibody (previously mAb7968), was derived from an anti-CFH autoantibody that is associated with an early stage, non-metastatic phenotype in non-small cell lung cancer (NSCLC) [14, 19, 20]. GT103 recognizes the same epitope as the anti-CFH autoantibody, induces CDC of tumor cells, and does not bind native, soluble CFH in blood. In mouse models, GT103 has anti-tumor growth activity with no effect on normal tissues. Here we show in vitro that GT103 binding to CFH on tumor-cell derived exosomes in the presence of complement results in their complement-dependent lysis. In addition, GT103 treated exosomes are subject to antibody-dependent phagocytosis.

## Materials and methods

### EV and exosome isolation

**Terminology.** In this report we designate vesicles isolated using an exosome ultracentrifugation protocol or a commercial exosome isolation kit as "EVs" as opposed to "exosomes" to allow for possible contamination with other types of vesicles. However, binding of EVs to anti-CD63 beads in order to facilitate flow cytometry provides an extra level of purification for exosomes for which CD63 is a specific marker [21]. In these particular experiments we designate these vesicles exosomes.

**EV isolation from human samples.** Duke University abides by the ethical principles outlined in the Belmont Report. The Duke University Health System Institutional Review Board approved this study and all subjects gave written informed consent for the use of their tissue. EVs were isolated from both early stage and late stage (i.e., metastatic) NSCLC patient plasma samples by ultracentrifugation. First, platelets were removed from plasma samples using low speed centrifugation, then EVs were isolated by ultracentrifugation of the supernatants in a Beckman SW41Ti rotor for 25 min at 35,000 rpm (210,000 x g) at 8 °C. The pellets were washed in 10 ml PBS, centrifuged again at 35,000 rpm, and the final pellet was resuspended in 100 µl PBS. Nanoparticle tracking analysis (NTA) of resuspended pellets from two early stage patient samples showed no EVs were present whereas NTA of pellet material from two late stage patient samples revealed EVs in the 30–350 nm size range. (The majority of these vesicles—~75%—were in the size range of exosomes and the remainder were larger, but as they were not purified further, we will term them EVs.)

**EV/exosome isolation from cell line conditioned media.** Cell lines were cultured in RPMI 1640 medium + 10% exosome-depleted fetal bovine serum (ThermoFisher Scientific) for 48 hr. Peripheral blood mononuclear cells (PBMCs) were isolated by Ficoll separation from the blood of a normal volunteer and were also cultured in this exosome depleted medium for 48 hr. Conditioned media were collected and EVs were isolated using the Total Exosome Isolation Kit (Invitrogen 4478359) according to the manufacturer's recommendations. To facilitate flow cytometry of human exosomes, EVs isolated using the kit were bound to human anti-CD63 magnetic Dynabeads (Invitrogen 10606D). For exosome conjugation to beads, the anti-CD63 Dynabeads were resuspended and washed twice with isolation kit buffer (PBS with 0.1% BSA). EV protein and beads were mixed in a ratio of 1 µg protein (as determined by BCA protein assay with BSA as standard): 1 µl beads and incubated at 4˚C on a rotator. Following this incubation, bead-bound exosomes were washed twice in isolation kit buffer. (Exosomes bound to anti-CD63 beads will be called "exosome-bead conjugates.")

## Antibody binding to exosomes

The NCI-H460 large cell lung carcinoma cell line was cultured in RPMI-1640 with 10% exosome-free fetal bovine serum, conditioned medium was collected, and EVs were isolated from the medium using the Total Exosome Isolation Kit and conjugated to anti-CD63 Dynabeads as described above. For each antibody binding reaction, an amount of exosome-bead conjugate containing 10 µg exosome protein was used. Exosome-bead conjugates were washed twice in 0.1% BSA in PBS. Primary antibody (GT103 or IgG subtype-matched negative control antibody 7B2) was added at 200 µg/ml and mixtures were incubated for 1 hr on ice. Exosome-bead conjugates were washed twice in 0.1% BSA in PBS. An anti-human Alexa Fluor-647-conjugated secondary antibody (Jackson Immunoresearch 109-605-003) diluted 1:100 was added, and mixtures were incubated 1 hr on ice. Exosome-bead conjugates were washed twice in 0.1% BSA in PBS and bead-bound fluorescence was analyzed on a BD FACSCanto flow cytometer.

## Complement-dependent lysis of exosomes

Exosome-bead conjugates containing 10 µg of exosome protein were labeled with 25 µM calcein AM (BD Biosciences, #564061) overnight at 37˚C. Exosome-beads were washed three times in wash buffer (PBS with 0.5% BSA) to remove unincorporated dye, and incubated with antibodies. In the human plasma exosome lysis experiment, GT103 and/or anti-CD59 mAb YTH53.1 (Santa Cruz Biotechnology) were used at 50 µg/ml. In the cell line exosome lysis experiments, GT103, a subtype-matched negative control antibody, or a Fab fragment of

GT103 [14, 22] were used at 200 μg/ml. Mixtures were incubated for 1 hr at 37˚C and washed twice in wash buffer. Following antibody treatment, exosome-bead conjugates were incubated with 10% normal human serum (NHS), heat-inactivated NHS, or specific complement factor-depleted serum for 1 hr at 37˚C and washed twice. (NHS and complement C1q, C4, and Factor B depleted sera were obtained from Complement Technology, Inc., Tyler, TX.) Beads were incubated with buffer containing 5% Triton X-100 for 10 minutes as a positive control for calcein release. Mixtures were analyzed for calcein release by flow cytometry on a FACSCanto flow cytometer (BD Biosciences). A decrease in signal indicates lysis.

### Antibody-dependent phagocytosis of exosomes

Macrophages were isolated by a published method [23]. Briefly, PMBCs from a human volunteer were plated in RPMI w/ L-Gln + 10% FBS, 8% normal human serum, 20 mM HEPES, 1% pen/strep and 50 ng/ml human macrophage colony stimulating factor (Cell Signaling 8929SC) for 5–6 days in order to induce macrophage differentiation. After washing with cold PBS, macrophages were harvested by gentle scraping of the plate. NCI-H460 EVs were isolated using Total Exosome Isolation Reagent and conjugated to anti-CD63 beads overnight at 4˚C to prepare exosome-bead conjugates as described above.

For the phagocytosis assay, exosome-bead conjugates were labeled with 5 μM carboxyfluorescein succinimidyl ester (CFSE; Biolegend 79898) for 20 min at 37 ᵒC then washed twice in 0.1% BSA in PBS. Macrophages were labeled with 5 μM Cell Trace Violet (Invitrogen C34571) for 20 min at 37 ᵒC then washed with five volumes of RPMI plus 10% FBS to quench the signal and suspended at $1x10^6$ cells/ml in RPMI. Cell Trace Violet-labelled macrophages (60 μl) and CFSE labeled exosome-bead conjugates (10 μg in 50 μl RPMI) were mixed in wells of a 96-well plate. GT103, or IgG control antibody (200 μg/ml) or no antibody was added to the exosome-bead conjugates and macrophages, in duplicate wells. The plate was centrifuged at 200 x g for 5 min to settle the contents of the wells and was incubated at 37˚C for 4 hours. Cells were detached with cold PBS and sorted in a BD FACSCanto flow cytometer, gating for doubly-labeled cells, indicating phagocytosis.

## Results

### CFH is present on EVs from tumor cell lines and patients with metastatic lung cancer

In order to examine the function of CFH in EVs, we made use of the anti-CFH antibody GT103. To prove that CFH is the target of GT103 in EVs, we probed a western blot of EVs purified from the conditioned media from the CMT167 murine lung cancer cell line and two CFH CRISPR/Cas9 knockout derivatives. GT103 recognized CFH in EVs derived from the wild type cell line and no protein in EVs derived from the knockout cell lines (**Fig 1**).

Next, we evaluated levels of CFH in a variety of tumor cell line EVs. We were particularly interested in whether levels of cell line-derived EVs correlate with metastatic potential. EVs were isolated from conditioned media using a commercial kit and were analyzed by western blot, probing with GT103. EVs from the highly metastatic murine cell lines B16-F10 [24] and Lewis lung cancer (LLC)-met (Dr. A.M. Pendergast, Duke University, personal communication) had substantially greater levels of CFH in equivalent amounts of EV protein than the less metastatic parental cell lines B16 and LLC respectively (**Fig 2A**). EVs derived from two human lung cancer cell lines, A549 and NCI-H460 also contained CFH. To demonstrate that GT103 binds to intact EVs, NCI-H460 EVs were conjugated to anti-CD63 beads. As CD63 is an exosome marker, the EVs that bind to the beads should be predominantly exosomes. The

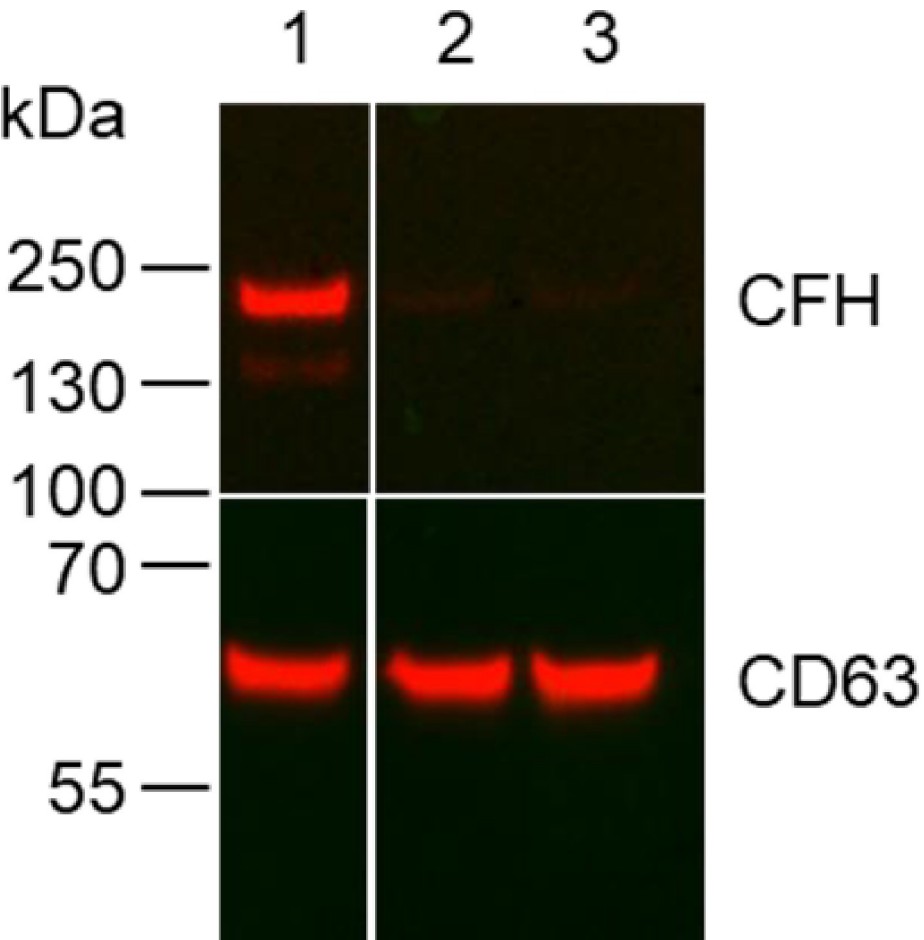

**Fig 1. CFH in EVs from CMT167 wild type vs. CFH knockout lung cancer cell lines.** EVs were isolated from cell line conditioned media and 7.5 μg protein were western blotted and probed with GT103 and an anti-human IgG-HRP, then stripped and probed with anti-CD63 (SBI Biosystems) and a goat-anti-rabbit-HRP conjugate. A composite of the two images is shown. Lanes contain EV protein from 1, wild type CMT167; 2 and 3, two different CFH-CRISPR/Cas9 knockout cell lines.

exosome-bead conjugates were incubated with GT103 or control antibody and subjected to flow cytometry. Binding of GT103 to the NCI-460 exosome-bead conjugates was significantly greater than binding of control antibody (**Fig 2B**). Exosomes were also isolated from normal human PBMCs. Binding of GT103 to PBMC exosome-bead conjugates was not significantly different than binding of control antibody to PBMC exosome-bead conjugates. In contrast, binding of GT103 to NCI-460 exosome-bead conjugates was significantly greater than binding of control antibody (**S1A Fig**).

We also wished to determine whether CFH in EVs correlates with metastasis in lung cancer patients. To this end, we isolated EVs by ultracentrifugation of plasma of 10 early stage (non-metastatic) lung cancer patients and 9 metastatic lung cancer patients, followed by western blot analysis of equal volumes of pellet material, probing with GT103. However, by NTA analysis, we found that most early stage patients had no detectible EVs. In contrast, NTA analysis showed that EVs were abundant in the plasma of metastatic patients. These EVs were associated with CFH (**Fig 3**).

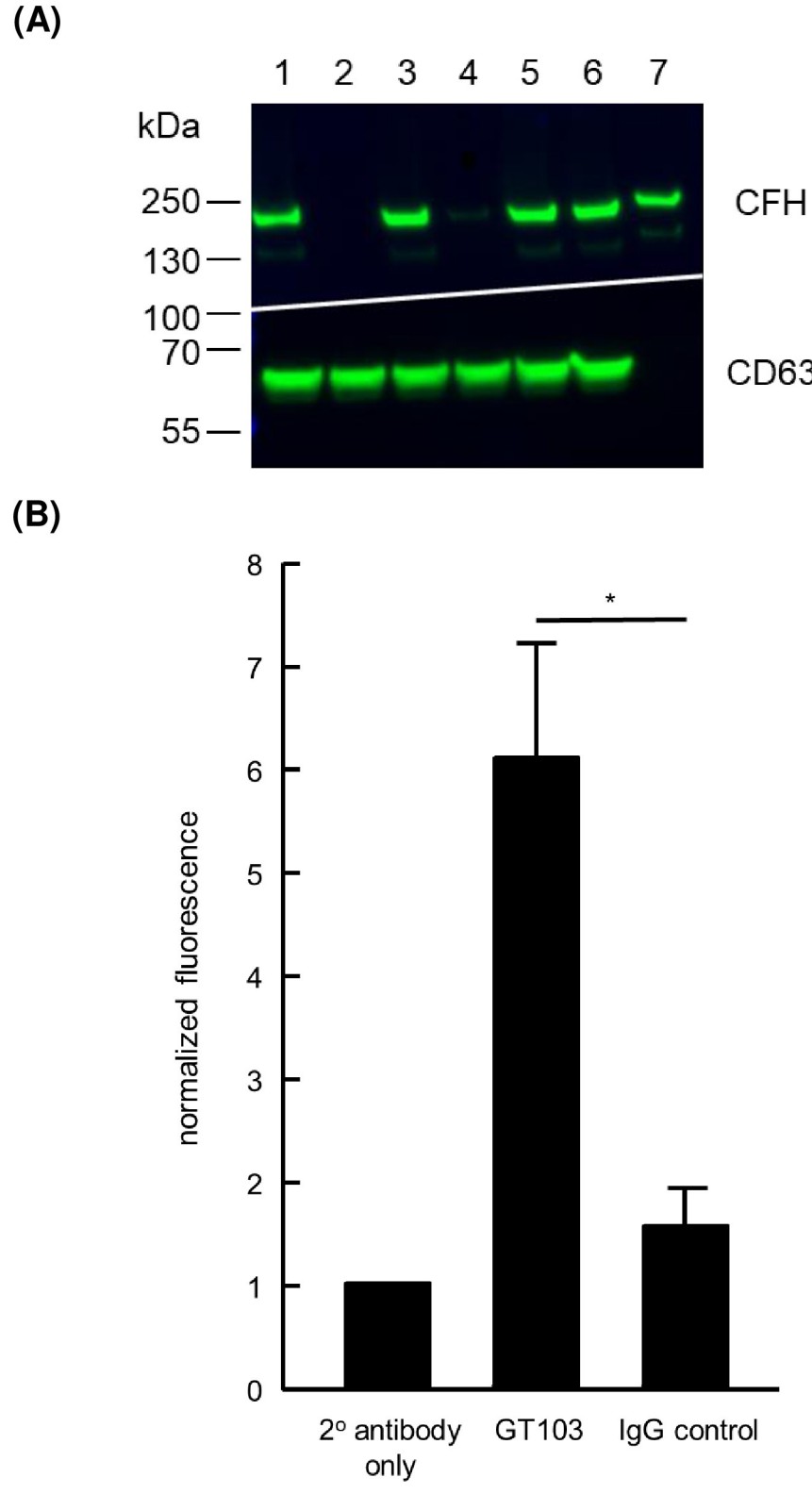

**Fig 2. GT103 binding to CFH in tumor cell line-derived EV protein and to intact tumor cell EVs.** Cell lines were cultured in exosome-free medium and EVs were isolated from conditioned medium with Total Exosome Isolation Reagent. **(2A) Western blot of EV protein probed with GT103 and anti-CD63.** 5 μg EV protein was loaded per lane. The blot was probed with 0.5 μg/ml GT103 and a secondary antibody-HRP conjugate, stripped, and reprobed with an

anti-CD63 and secondary antibody-HRP conjugate as described in the legend to Fig 1. A composite of the two images is shown. Lanes contain EV protein from the following cell lines: 1, B16F10; 2, B16; 3, LLC-met; 4, LLC 5, A549; 6, NCI-H460. Lane 7 contains 25 ng purified human CFH (Complement Technology, Inc., Tyler, TX). **(2B) Flow cytometry of antibody-bound NCI-H460-derived exosomes.** Tumor cell EVs were conjugated to CD63 Dynabeads. A quantity of beads containing 10 μg EV protein was incubated sequentially with GT103 or an IgG-subtype matched negative control antibody (200 μg/ml) and an anti-human Alexa Fluor-647-conjugated secondary antibody (1:100). Bead bound fluorescence was analyzed on a BD FACSCanto flow cytometer. Each reaction was performed in duplicate and the experiment was repeated three times. Mean fluorescence in each experiment was normalized to a secondary antibody alone control. The normalized means from the GT103 and IgG treatments from the three experiments were used to calculate overall mean, SD, and P value for the GT103 vs. IgG comparison. These data calculations are provided in the S1 Dataset on the tab for Fig 2. Significance was assessed, here and in subsequent experiments, using Student's t-test and all exact P values are provided in the Dataset. *P<0.05.

## Anti-CFH and anti-CD59 antibodies enable complement-dependent lysis of tumor-derived exosomes

We hypothesized that GT103 could destroy exosomes by a mechanism analogous to CDC. A previous study had shown that neutralization of the membrane complement regulatory proteins CD55 and CD59, in the presence of serum and a "sensitizing" antibody (i.e., a classical pathway initiating antibody) caused complement lysis of exosomes derived from a B lymphoblastoid cell line [25]. Here we asked if neutralization of CFH with or without neutralization of CD59 causes complement lysis of exosomes. We conjugated exosomes from a late stage NSCLC patient to CD63-beads and labeled them with calcein. We then added antibodies against CFH (GT103), CD59 (YTH53.1, Santa Cruz Biotechnology), CFH+CD59, or control IgG, plus NHS as a source of complement. We performed flow cytometry on the exosome-

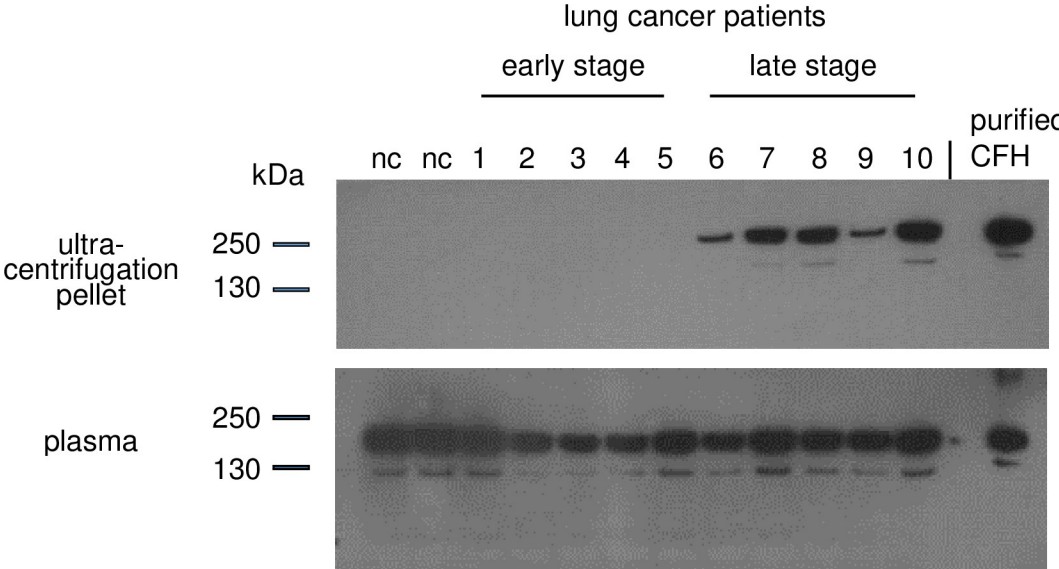

**Fig 3. CFH in ultracentrifugation pellet and plasma of lung cancer patients.** EVs were isolated from plasma by ultracentrifugation. Equal volumes of ultracentrifugation pellet and original plasma from each patient were subjected to western blot analysis. The blots were each probed with GT103 followed by an anti-mouse IgG-HRP secondary reagent (to prevent background binding of a human secondary reagent to human IgG in the samples). Top blot: The pellet from 5 early stage (1–5) and 5 late stage (6–10) patients. The last lane contains 20 ng purified human CFH (Complement Technology, Inc.) Bottom blot: Plasma from the same patients. The pellet and plasma from two control patients with no cancer (nc) are also included on the blot. Shown are results from a subset of 10 lung cancer patients of a total of 19 analyzed; only one early stage patient had a detectible CFH band in EVs. (See S2 Fig for 9 additional human samples, including the early stage patient with CFH-containing EVs).

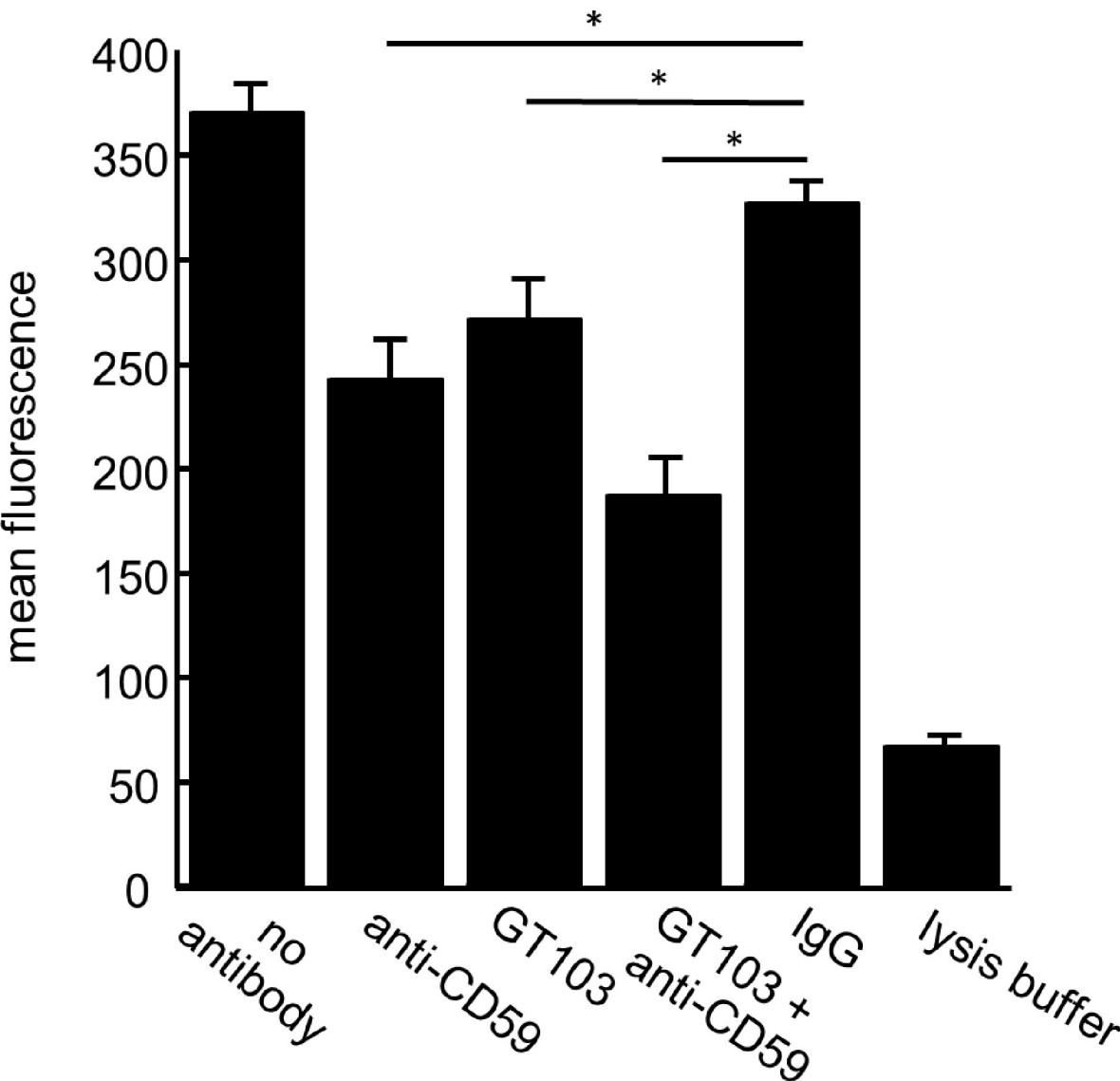

**Fig 4. Lysis of human exosomes after neutralizing complement protective proteins.** EVs were isolated from the serum of an NSCLC patient using the ExoQuick Ultra isolation kit (System Biosciences). Exosomes were conjugated to CD63 beads and labeled with 25 μM calcein AM. Exosomes were treated with 50 μg/ml antibodies in triplicate followed by treatment with 10% NHS as a source of complement for 1 hr at 37 °C. Mixtures were analyzed by flow cytometry. Data are represented as mean +/- SD. *P<0.05.

conjugated beads; loss of label is indicative of lysis. We found that compared to control IgG, antibodies against CFH, CD59, and CFH + CD59 increased lysis by 17%, 26%, and 43%, respectively (Fig 4); however, only in the case of dual antibody treatment was this increase significant.

We found that exosomes from the human lung cancer cell line NCI-H460 were much more susceptible to GT103-mediated lysis than the exosomes from the single patient in the preceding experiment. We used NCI-H460 exosomes to ask whether GT103-mediated exosome lysis occurs by the alternative complement pathway (initiated when CFH is blocked by GT103) or by the classical pathway (in which the GT103 antibody itself is the "sensitizing" antibody). We treated calcein-labelled NCI-H460 exosomes with GT103 or negative control IgG in the presence of NHS or heat-inactivated NHS (as a control for complement dependence). GT103

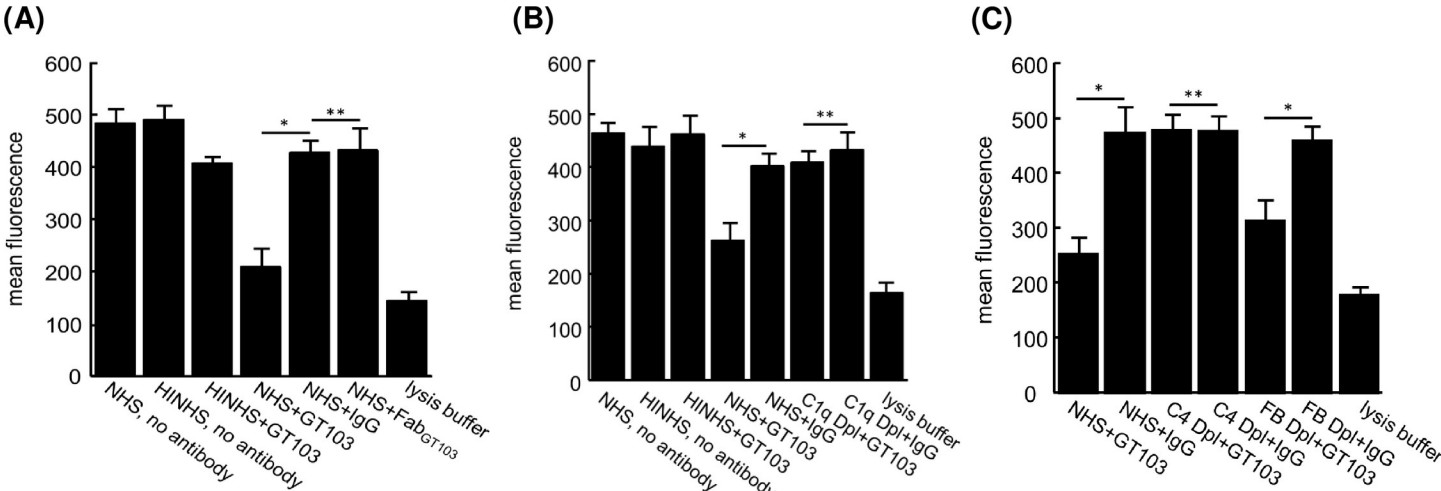

**Fig 5. GT103 mediates complement lysis of exosomes by the classical pathway.** Exosomes isolated from the conditioned medium of NCI-H460 cells were conjugated to anti-CD63 beads and labeled with calcein. Antibodies were added at 200 μg/ml. Mixtures were incubated for 1 hr on ice, then 10% normal human serum (NHS), heat-inactivated serum (HINHS), or a complement factor-depleted serum was added. After incubation for 1 hr at 37˚C calcein fluorescence was analyzed by flow cytometry. As a positive control, exosome-bead conjugates were treated with lysis buffer for 10 min. Each condition was performed in triplicate. Exosome lysis is indicated by a reduction in fluorescence. **(A)** Lysis in the presence of GT103 or GT103-derived Fab; **(B)** Lysis in the presence of NHS or C1q depleted serum; **(C)** Lysis in the presence of NHS or C4 or Factor B depleted serum Data are represented as mean +/- SD.*P<0.05; **P>0.05.

significantly increased exosome lysis over the negative control antibody in a complement-dependent manner (**Fig 5A–5C**). A Fab fragment of GT103 was unable to substitute for the full-length antibody (**Fig 5A**), indicating that the Fc portion was required for exosome lysis, and that lysis proceeded along the classical complement pathway. This conclusion was confirmed using depleted NHS in the lysis assay: When the classical pathway was eliminated using C4 or C1q depleted serum, no significantly significant exosome lysis was seen (**Fig 5B and 5C**). In contrast, when the alternative pathway was blocked using Factor B depleted serum, GT103-mediated exosome lysis still occurred, although to a reduced extent compared to undepleted serum (**Fig 5C**).

In addition, in a complement-dependent exosome lysis experiment carried out in the presence of NHS, GT103 had no effect on the lysis of normal PBMC exosomes conjugated to anti-CD63 beads. In contrast, GT103 lysed NCI-H460 exosomes conjugated to anti-CD63 beads (**S1B Fig**).

### The anti-CFH antibody enables phagocytosis of tumor cell-derived exosomes

Upon classical pathway activation, immune complexes are formed on the target cell surface [26]. Therefore, we hypothesized that GT103-treated exosomes might be opsonized by immune complexes, and therefore could be targeted by phagocytes bearing Fc receptors. Exosomes from NCI-H460 cells and normal human macrophages were differentially labeled, mixed, and incubated with GT103 or control antibody. Flow cytometry was used to detect doubly positive cells indicative of phagocytosis. We observed a statistically significant increase in phagocytosis of exosomes treated with GT103 vs. those treated with the control IgG (**Fig 6**).

### Discussion

Monoclonal antibody therapies that target tumor cells not only have cytotoxic or cytostatic effects but can bring about immune system activation [27]. By binding to cell surface receptors,

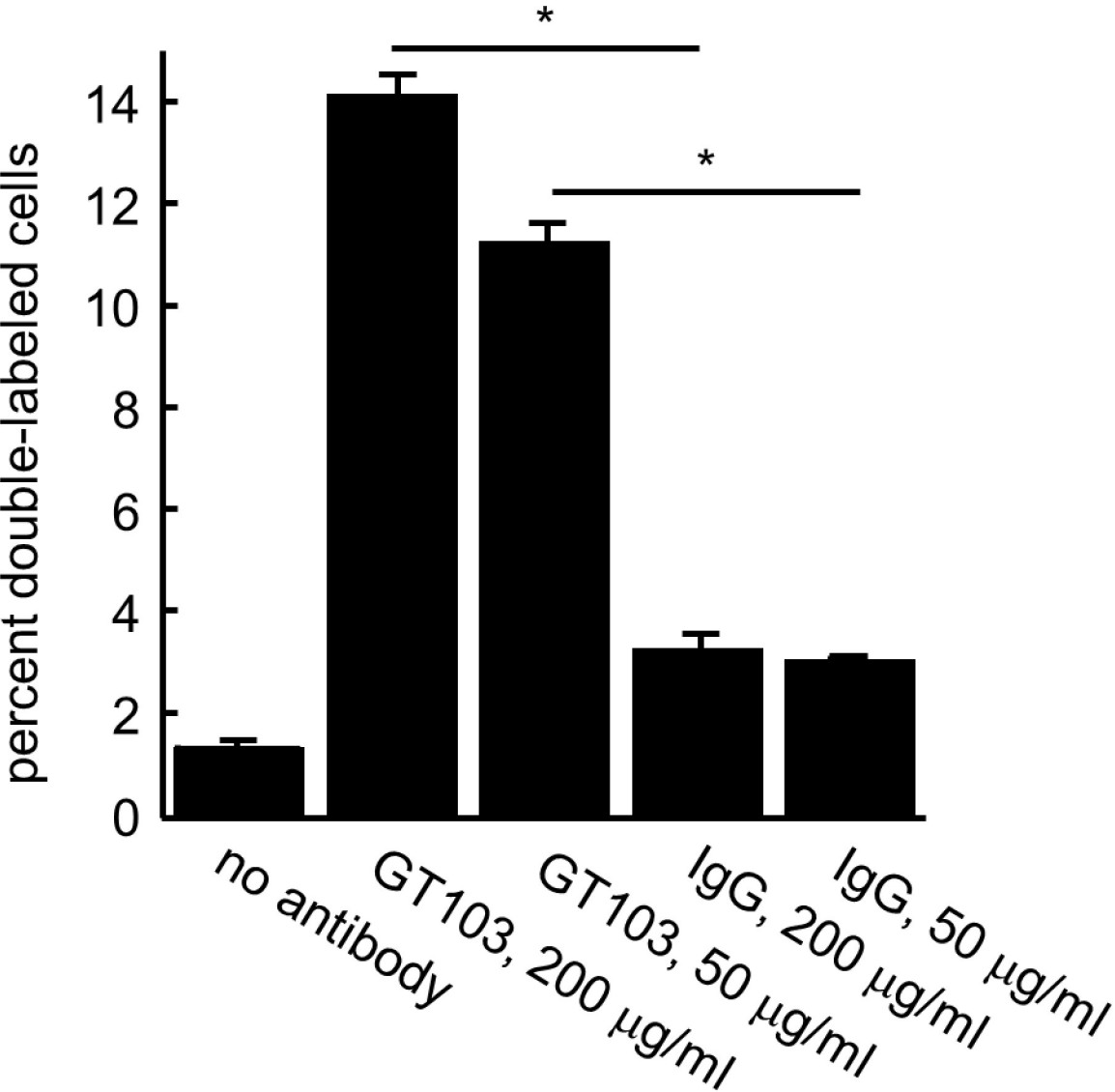

**Fig 6. GT103-mediated phagocytosis of NCI-H460 exosomes.** Exosomes from the NCI-H460 large cell lung carcinoma cell line were conjugated to anti-CD63 beads then exosome-bead conjugates were labeled with CFSE. Macrophages (isolated from normal PBMCs) were labeled with Cell Trace Violet. CFSE-labeled exosome-bead conjugates and Cell Trace Violet-labelled macrophages were mixed and either no antibody, GT103 or IgG control antibody (at 200 μg/ml or 50 μg/ml) was added to the mixtures in duplicate. After incubation at 37°C for 4 hours, cells were sorted by flow cytometry, gating for doubly-labeled cells. Data are represented as mean +/- SD. *P<0.05.

these antibodies exert their effects by one or more mechanisms [28, 29] including those that directly impact receptor signaling (leading to growth inhibition or induction of apoptosis), and those that promote innate immune responses (such as antibody-dependent or complement-dependent cytotoxicity and phagocytosis). Although the effect of many antibody therapies on tumor cells is well characterized, their effect on tumor cell-derived EVs or exosomes has not been studied. Although the EV is not a cell, therapeutic antibodies could, in theory, bind to their cognate receptors incorporated into the EV membrane and generally alter the immunogenicity of the EV by promoting antibody-dependent immune processes. Antibodies could also block specific EV functions, such as the ability of the EV to dock at its intended destination and deliver cargo, e.g., if the antibody blocks integrins [30, 31], or the ability of the EV

to suppress T cells via PD-L1 [10, 11]. This is an important area of study as EVs/exosomes have roles in tumor promotion and development of metastasis [9]. Lowering the level or blocking the functionality of exosomes in advanced cancer might be a useful therapeutic strategy.

We have used GT103 as a tool for probing the vulnerability of exosomes to destruction by innate immune mechanisms. GT103 is a fully human anti-CFH monoclonal antibody currently in a Phase Ib clinical trial for advanced lung cancer (clinicaltrials.gov). GT103 acts through CFH to kill tumor cells by promoting complement activation, culminating in the formation of the lytic MAC [14, 20]. In vitro, the GT103 antibody operates predominantly by the classical pathway (**S2 Fig**); the classical pathway is initiated by the formation of immune complexes [26]. However, to a lesser extent, the antibody also promotes the alternative pathway, initiated by the alternative convertase C3bBb, by competing for the CFH epitope that binds C3b in the SCR19 domain [14, 20]. In vivo GT103 inhibits tumor growth and stimulates lymphocytic infiltration of tumors. Since GT103 mediates tumor cell destruction, here we asked if it could act through CFH to destroy tumor cell-derived exosomes.

Others have shown that exosomes/EVs determine the organotropism of metastasis and the preparation of a premetastatic niche [30, 32, 33]. Therefore, we asked whether CFH level in EVs correlates with metastatic potential of cell lines that produce them. After demonstrating that CFH is the target of GT103 in EVs, we surveyed several tumor cell lines and found that CFH was more prevalent in the EVs of the metastatic variants of B16 and LLC than the parental cell lines. In HCC, highly metastatic cell lines were also more enriched in CFH than less metastatic cell lines [13]. EVs from metastatic HCC cell lines promoted metastasis that was abrogated by treatment with GT103, providing evidence that CFH preserves the metastasis-promoting function of EVs. We also found that CFH-containing EVs were present in the plasma of patients with metastatic NSCLC but were not detected in the plasma of most patients with early stage, non-metastatic NSCLC. This contrasts with findings in melanoma, where exosome number did not differ with clinical stage; however, exosome protein concentration was higher in stage 4 melanoma compared to other stages [33]. Our experiment has the limitation that we were unable to detect EVs in the plasma of early stage patients so we were unable to compare the relative abundance of CFH per exosome in early vs. late stage NSCLC.

In order to examine whether CFH has a protective function in exosomes, we isolated exosomes from the plasma of a patient with metastatic NSCLC and found that they were lysed by GT103 in the presence of complement, and the degree of lysis could be increased by addition of an antibody to CD59. CD59 and CFH protect cells from lysis by different mechanisms although it is not known if these mechanisms operate the same way on exosomes as they do on cells. On cells, CFH is transiently associated with the membrane via its association with proteoaminoglycans and prevents an early step in MAC formation–deposition of C3b–while CD59 is stably associated with the membrane via a glycophosphatidylinositol anchor and prevents a late step in MAC formation, inhibiting the insertion of C9 into the terminal complex [34–36]. This experiment demonstrates that EVs can be made vulnerable to destruction by antibodies to complement regulatory proteins.

In order to delve into the mechanism of lysis further, we examined the ability of GT103 to lyse exosomes from the NCI-H460 lung cancer cell line. We found that GT103-mediated exosome lysis proceeded predominantly by the classical pathway, and to a lesser extent by the alternative pathway, because lysis did not occur with a Fab fragment of GT103, nor in C1q or C4 depleted serum, but was only slightly reduced in Factor B depleted serum. Furthermore, GT103 promoted phagocytosis of exosomes, presumably due to recognition of immune complexes on the exosome surface by macrophage Fc receptors.

In this report, we have demonstrated that exosomes are vulnerable to destruction by complement lysis and phagocytosis. Although this work was performed in vitro, it should stimulate

further investigation into exosome stability and activity in vivo, and provide ideas about how exosome efficacy in promoting metastasis and immunosuppression might be manipulated as a novel therapeutic strategy.

## Supporting information

**S1 Dataset.**
(XLSX)

**S1 Raw images.**
(PDF)

**S1 Fig. GT103 binding to, and lysis of, PBMC-derived and NCI-H460 tumor cell-derived exosomes.** PBMCs isolated from the blood of a normal volunteer and the NCI-460 lung tumor cell line were each cultured in exosome free medium. EVs were isolated from the conditioned media using the Total Exosome Isolation Kit (Invitrogen 4478359) and conjugated to anti-CD63 beads (Invitrogen 10606D). **(A)** Antibody binding to exosomes. Binding of GT103 or control IgG to the exosome-bead conjugates was measured by flow cytometry. **(B)** Antibody-mediated lysis of exosomes. Lysis of exosomes in exosome-bead conjugates in the presence of GT103 or IgG and NHS as a source of complement was measured by flow cytometry in a calcein release assay in which loss of label is indicative of lysis. In both experiments, reactions were run in duplicate. Data are represented as mean +/- SD; significance was assessed using Student's t-test. $^{*}$P<0.05; $^{**}$P>0.05.
(PDF)

**S2 Fig. GT103 western blot film of extracellular vesicles from 9 additional lung cancer patients.** EVs were isolated from patient plasma by ultracentrifugation and subjected to western blot analysis for CFH. The blot was probed with human GT103 as primary antibody followed by an anti-human-HRP secondary antibody-conjugate. Samples from early stage lung cancer patients are in lanes labeled 1–5 (histotype denoted in black), a sample from a control patient with no cancer (nc) is in the next lane, and samples from late stage lung cancer patients are in lanes labeled 6–9 (histotype denoted in red). Note lane 5 contains CFH-positive EVs from an early stage lung cancer patient mentioned in the text.
(PDF)

**S3 Fig. GT103-mediated complement-dependent cytotoxicity (CDC) of human lung cancer cells in the presence of normal or complement depleted sera.** A549 human lung cancer cells were incubated with no antibody, GT103, or IgG negative control in the presence of intact normal human serum (NHS), or serum depleted (Dpl) of Factor B (FB), C1q, or C4. After 24 hrs, lysis was measured by lactose dehydrogenase release using the CytoTox 96Ⓡ Non-Radio-active Cytotoxicity Assay (Promega, Madison, WI) according to the manufacturer's instructions, and expressed as percent cytotoxicity. In addition, cells incubated with heat inactivated NHS (HI-NHS) were included as a control for spontaneous lysis of cells occurring in serum with no complement or antibody; cells incubated with NHS were included as a control for spontaneous CDC in serum with no antibody. All reactions were run in triplicate. Data are represented as mean +/- SD; significance was assessed using Student's t-test. $^{*}$P<0.05.
(PDF)

## Acknowledgments

We thank Jacob Hoj and Ann Marie Pendergast of Duke University for the generous gift of the LLC-met cell line.

## Author Contributions

**Conceptualization:** Ryan T. Bushey, Elizabeth B. Gottlin, Michael J. Campa, Edward F. Patz, Jr.

**Data curation:** Ryan T. Bushey, Michael J. Campa.

**Formal analysis:** Ryan T. Bushey, Michael J. Campa.

**Investigation:** Ryan T. Bushey, Michael J. Campa.

**Methodology:** Ryan T. Bushey, Michael J. Campa.

**Project administration:** Edward F. Patz, Jr.

**Validation:** Ryan T. Bushey, Michael J. Campa.

**Writing – original draft:** Elizabeth B. Gottlin.

**Writing – review & editing:** Ryan T. Bushey, Elizabeth B. Gottlin, Michael J. Campa, Edward F. Patz, Jr.

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
