## [Decision Letter · Decision Letter 0]

14 Apr 2021

PONE-D-21-08454

Complement Factor H Protects Tumor Cell-Derived Exosomes from Complement-Dependent Lysis and Phagocytosis

PLOS ONE

Dear Dr. Patz,

Thank you for submitting your manuscript to PLOS ONE. After careful consideration, we feel that it has merit but does not fully meet PLOS ONE’s publication criteria as it currently stands. 

The comments of our two expert Reviewers are enclosed for your information.

We are willing to consider a revised version of your manuscript for publication, but it is mandatory that all criticisms raised during the review process will be adequately addressed.

Please submit your revised manuscript within one month. If you will need more time than this to complete your revisions, please reply to this message or contact the journal office at plosone@plos.org. Please include the following items when submitting your revised manuscript:

We look forward to receiving your revised manuscript.

Kind regards,

Marco Trerotola

Academic Editor

PLOS ONE

Journal Requirements:

I have read the journal's policy and the authors of this manuscript have the following competing interests: Drs. Gottlin, Campa, and Patz are founders of Grid Therapeutics, LLC, which is supplying GT103 for a Phase 1b clinical trial.

Reviewers' comments:

Reviewer's Responses to Questions

**Comments to the Author**

1. Is the manuscript technically sound, and do the data support the conclusions?

Reviewer #1: No

Reviewer #2: Yes

2. Has the statistical analysis been performed appropriately and rigorously? 

Reviewer #1: No

Reviewer #2: I Don't Know

3. Have the authors made all data underlying the findings in their manuscript fully available?

Reviewer #1: Yes

Reviewer #2: Yes

4. Is the manuscript presented in an intelligible fashion and written in standard English?

Reviewer #1: Yes

Reviewer #2: Yes

5. Review Comments to the Author

Reviewer #1: The current study of Bushey et al. is interesting and adds new insight into the role of complement regulation in tumor cell derived EVs. By using flow cytometric and immunological methods, Bushey et al. show how tumor cell EVs are protected from complement attack through binding of factor H. Importantly, factor H mediated evasion of complement by tumor cell EVs can be circumvented using a monoclonal antibody, GT103.

The manuscript is very well and clearly written but also raise some concerns mainly in the data presentation, statistical measurements and conclusions.

Major concerns:

1. Some experiments do not include appropriate replications. The statistics are not described in sufficient detail. The statistics used to measure significance should be mentioned in the methods section or in the figure legend. Please, specify whether variation in the Figures is shown as standard deviation.

a. Fig. 2: “. The experiment was performed three times and a representative result is shown”. Statistics cannot be calculated from one representative experiment. Please include replicates in the figure and calculate the p-value and variation from the three experiments.

b. Fig. 4. Fig. 5. Fig. 6. Is variation of the replicates indicated as SD?

2. Conclusions are not completely supported by the data:

a. “Anti-CFH antibody can be used to target tumor-derived exosomes for exosome destruction via innate immune mechanisms. These findings suggest that a therapeutic CFH antibody has the potential to inhibit tumor progression and reduce metastasis promoted by exosomes”. By studying only on tumor cell derived EVs does not rule out the possibility, that GT103 could also target EVs from “healthy cells”, or early stage EVs.

b. “Higher levels of CFH-containing EVs correlated with higher metastatic potential of cell lines.”. In Fig. 3. This is unclear. The pellet of early stage does not contain EVs? If the authors did not detect EVs in plasma of early stage patients how can this conclusion be made?

Minor concerns:

- Fig. 3. “Only one early stage patient had a detectible CFH band in EVs”. I assume this data is not shown. This should be included in the manuscript or supplementary data.

- Line 19: Here we show that….

- Fig.3. Control CFH shown in WB. Manufacturer? Please, include in methods.

- 145: “or a Fab fragment of GT103 were used at 200 µg/ml” add reference to a method.

- 147: exosome-bead conjugates?

- 160: NCl-H460 EVs were conjugated to CD63+ beads to prepare exosome-bead conjugates?

- 181: “GT103 recognized a protein produced by the wild type cell line with the mass of CFH and this band was missing in the knockout cell lines”. If not sure include a FH control in the WB or mention that GT103 recognized FH.

- 200: Binding of GT103 to the exosome-bead conjugates was significantly greater than binding to control antibody > binding of?

- 317: “Upon classical pathway activation, immune complexes are formed on the target cell surface”. Immune complexes are known to activate the classical pathway but if this true please, include a reference.

- “In vitro, the antibody operates predominantly by the classical pathway, initiated by the formation of immune complexes (unpublished data).” According to journal guidelines, unpublished work should not be cited. The data can be included as supplementary material or deposit the data in a publicly available database.

- 347-350: Please, include references

- Fig. 3. Why there is no CFH in early-stage EVs? Did the authors detect early-stage EVs? If they do not bind FH do they express less sialic acids or another ligand? The presence of sialic acids would be very easy to check in flow cytometry by for example using fluorescent labeled Maackia Amurensis Lectin I (MAL I). This should be discussed in the manuscript.

Reviewer #2: This paper by Bushey et al. examines whether EVs from cancer cells express factor H on their surface, and whether an antibody that targets factor h causes the EVs to be lysed. The manuscript is well clearly written and the data clearly presented. The data is all in vitro, which limits the insights into what effect this treatment would have on cancer in vivo. It is strengthened, though, by examination of samples from cancer patients. I have the following questions.

1. The phagocytosis assay studies exosome-bead conjugates. It does not seem that this assay will reflect what happens to free exosomes.

2. Figure 2 – the authors find greater factor H on EVs from metastatic cancer cell lines than less metastatic cells. Presumably the factor H came from the parent cells themselves. But since the EVs will be surrounded by plasma FH. But perhaps EVs from the other types of cancer would have factor H on them as soon as they are exposed to serum. The authors could test this, then expose the EVs to serum, wash, and see if any factor H bound. The question seems to be whether the cancer cell produces factor H, or whether there is some other compositional difference of the EVs that makes factor H stick to them.

3. The study found very few EVs in samples from early stage cancer patients. Some studes have reported high numbers of exosomes in normal patient plasma (e.g. Huang et al. BMC Genomics 2013;14:319. ) Is there something different about the isolation process that accounts for this? The authors used a commercial column to purify the EVs, but perhaps some characterization of the EVs as outlined in Théry, et al., 2018; DOI: 10.1080/20013078.2018.1535750 would he useful.

4. In figure 3 could probe the blot for CD63 and see if any exosomes were isolated from the early stage samples.

5. Lysis of the exosomes seems to involve the classical pathway. This suggests that blocking factor H function is less important than binding to an EV surface protein. Factor H has a fairly low affinity for most surfaces, so perhaps a transmembrane protein would be a better target.

6. Are the authors certain that the antibody does not bind to factor H on EVs from other non-cancer cell types? They could test binding to EVs isolated from other cell lines (e.g. macrophages). The authors have evidence that the antibody is specific for factor H on cancer cells, but if it did bind to factor H on healthy cell-EVs it could disrupt physiologic processes.

6. PLOS authors have the option to publish the peer review history of their article (what does this mean?). If published, this will include your full peer review and any attached files.

Reviewer #1: **Yes: **Dr. Karita Haapasalo

Reviewer #2: No

---

## [Author Response · Author response to Decision Letter 0]

12 May 2021

To the Editor and Reviewers:

Thank you for your comments on our manuscript, entitled “Complement Factor H Protects Tumor Cell-Derived Exosomes from Complement-Dependent Lysis and Phagocytosis” (PONE-D-21-08454). We have done additional experiments, added new figures as Supporting Information, revised the manuscript, and added references. In addition, we are supplying a raw western blot images file and a dataset file. Here we address each critique of the review point-by-point. In one copy of the revised manuscript, Tracked Changes show the revisions, except where references were added, so as to not cause excessive formatting changes in the document. These new references are highlighted in yellow in the revised manuscript and the line numbers where they are located are noted below. 

Journal Requirements:

All appears to be in order.

2. PLOS ONE now requires that authors provide the original uncropped and unadjusted images underlying all blot or gel results reported in a submission’s figures or Supporting Information files. When you submit your revised manuscript, please ensure that your figures adhere fully to these guidelines and provide the original underlying images for all blot or gel data reported in your submission.

A file with unaltered blot images called S1_raw_images has been submitted as Supporting Information.

I have read the journal's policy and the authors of this manuscript have the following competing interests: Drs. Gottlin, Campa, and Patz are founders of Grid Therapeutics, LLC, which is supplying GT103 for a Phase 1b clinical trial.

Please confirm that this does not alter your adherence to all PLOS ONE policies on sharing data and materials, by including the following statement: "This does not alter our adherence to PLOS ONE policies on sharing data and materials.” (as detailed online in our guide for authors. If there are restrictions on sharing of data and/or materials, please state these. Please note that we cannot proceed with consideration of your article until this information has been declared.

Please know it is PLOS ONE policy for corresponding authors to declare, on behalf of all authors, all potential competing interests for the purposes of transparency. PLOS defines a competing interest as anything that interferes with, or could reasonably be perceived as interfering with, the full and objective presentation, peer review, editorial decision-making, or publication of research or non-research articles submitted to one of the journals. Competing interests can be financial or non-financial, professional, or personal. Competing interests can arise in relationship to an organization or another person.

We confirm this will not alter our adherence to PLOS ONE policies. The sentence “This does not alter our adherence to PLOS ONE policies on sharing data and materials” has been added to the Competing Interests statement in the manuscript and the full Competing Interests statement is also present in the revised cover letter.

We have added references as requested by the reviewers. We have checked every reference in the list to make sure it is correct and has not been retracted.

Reviewers' comments:

Reviewer's Responses to Questions

Comments to the Author

1. Is the manuscript technically sound, and do the data support the conclusions?

Reviewer #1: No

Reviewer #2: Yes

2. Has the statistical analysis been performed appropriately and rigorously?

Reviewer #1: No

Reviewer #2: I Don't Know

3. Have the authors made all data underlying the findings in their manuscript fully available?

The requires authors to make all data underlying the findings described in their manuscript fully available without restriction, with rare exception (please refer to the Data Availability Statement in the manuscript PDF file). The data should be provided as part of the manuscript or its supporting information, or deposited to a public repository. For example, in addition to summary statistics, the data points behind means, medians and variance measures should be available. If there are restrictions on publicly sharing data—e.g. participant privacy or use of data from a third party—those must be specified.

Reviewer #1: Yes

Reviewer #2: Yes

4. Is the manuscript presented in an intelligible fashion and written in standard English?

Reviewer #1: Yes

Reviewer #2: Yes

5. Review Comments to the Author

Reviewer #1: The current study of Bushey et al. is interesting and adds new insight into the role of complement regulation in tumor cell derived EVs. By using flow cytometric and immunological methods, Bushey et al. show how tumor cell EVs are protected from complement attack through binding of factor H. Importantly, factor H mediated evasion of complement by tumor cell EVs can be circumvented using a monoclonal antibody, GT103.

The manuscript is very well and clearly written but also raise some concerns mainly in the data presentation, statistical measurements and conclusions.

Major concerns:

1. Some experiments do not include appropriate replications. The statistics are not described in sufficient detail. The statistics used to measure significance should be mentioned in the methods section or in the figure legend. Please, specify whether variation in the Figures is shown as standard deviation.

The figure legends of every bar graph now include the statement: “Data are represented as mean +/- SD.” This is followed by the P value (designated as less than or greater than 0.05). (Actual P values are given in the S1_Dataset attachment.) The legend of the first bar graph, Fig 2B, also states that significance in this and subsequent figures was assessed using Student’s t-test. Every figure legend contains the number of replicates that went into the mean.

The experiment showing binding of GT103 to NCI-H460 tumor cell line exosomes was performed three times (Fig. 2B). (This binding experiment was also repeated as a positive control in the new S1A Fig.) The experiment showing GT103-mediated lysis of these exosomes was performed twice (Fig. 5 and as a positive control in the new S1B Fig.) The experiment showing GT103-mediated phagocytosis of these exosomes was performed once, but at two different concentrations of antibody (Fig. 6).

a. Fig. 2: “. The experiment was performed three times and a representative result is shown”. Statistics cannot be calculated from one representative experiment. 

Please include replicates in the figure and calculate the p-value and variation from the three experiments.

For Fig 2B, we interpreted the reviewer’s comment to mean that the data from the original and two replicate experiments should be combined into one figure with associated means, SDs, and a P value for the significance of the difference between the GT103 and control IgG treatments. Therefore, we have combined all the replicate data points from the three experiments, with associated statistics, into a new Fig. 2B.

b. Fig. 4. Fig. 5. Fig. 6. Is variation of the replicates indicated as SD? 

Yes. As stated above, we have added this explanation.

2. Conclusions are not completely supported by the data:

a. “Anti-CFH antibody can be used to target tumor-derived exosomes for exosome destruction via innate immune mechanisms. These findings suggest that a therapeutic CFH antibody has the potential to inhibit tumor progression and reduce metastasis promoted by exosomes”. By studying only on tumor cell derived EVs does not rule out the possibility, that GT103 could also target EVs from “healthy cells”, or early stage EVs.

In the manuscript we referenced evidence from our prior publications that the GT103 antibody appears to have tumor cell specificity. Since exosomes are not cells, we understand the concern about whether GT103 shows tumor exosome specificity. Therefore, we tested GT103 binding to, and lysis of, exosomes from normal PBMCs in parallel with exosomes from lung tumor cell line NCI-460. In the GT103 exosome binding experiment, we found that GT103 binds to NCI-460 exosomes but not to normal PBMC exosomes (new S1A Fig). In the GT103 exosome lysis experiment, we found that GT103 causes lysis of NCI-H460 exosomes but not normal PBMC exosomes (new S1B Fig).

b. “Higher levels of CFH-containing EVs correlated with higher metastatic potential of cell lines.”. In Fig. 3. This is unclear. The pellet of early stage does not contain EVs? If the authors did not detect EVs in plasma of early stage patients how can this conclusion be made?

The quoted sentence excerpted by the reviewer from the Abstract applies to EVs from the cell lines in Fig. 2A, not EVs from lung cancer patients. Fig. 2A shows side-by-side comparison of the levels of CFH-containing EVs from B16/B16F10 cells and LLC/LLC-met cells. Each of these pairs is composed of a parental cell line with low metastatic potential and a derivative with higher metastatic potential. In order to support the phenotypes of the cell lines, we supplied the original literature reference for B16/B16F10 and a letter of support from Dr. Pendergast for LLC/LLC-met.

For the experiment of Fig. 3 that shows the CFH content of EVs from lung cancer patients, we originally hypothesized that EVs from late stage patients would contain more CFH than the EVs from early stage patients. However, we were unable to answer this question because we could not isolate enough EVs from the early stage patients. We stated this limitation of the experiment in the Discussion. The reason we are including this study in this paper at all is because (1) Nanoparticle Tracking Analysis (NTA) showed that total EVs are more abundant in late stage than early stage lung cancer and (2) the western blot of Fig. 3 shows that EVs from late stage patients contain CFH. We think both findings are relevant.

Minor concerns:

- Fig. 3. “Only one early stage patient had a detectible CFH band in EVs”. I assume this data is not shown. This should be included in the manuscript or supplementary data.

Data from 9 additional lung cancer patients, including the one early stage patient with detectible CFH in EVs, are now shown in S2 Fig. 

- Line 19: Here we show that…. Added the word “that”.

- Fig.3. Control CFH shown in WB. Manufacturer? Please, include in methods.

We added this information to all the relevant figure legends since there is not a section for western blots in the Methods.

- 145: “or a Fab fragment of GT103 were used at 200 µg/ml” add reference to a method. We added two references to the generation of this Fab and its crystal structure (revision line 157). 

- 147: exosome-bead conjugates? Changed “exosome-beads” to “exosome bead conjugates”.

- 160: NCl-H460 EVs were conjugated to CD63+ beads to prepare exosome-bead conjugates? We changed the wording as per your suggestion. (Also changed the CD63+ terminology to anti-CD63.)

- 181: “GT103 recognized a protein produced by the wild type cell line with the mass of CFH and this band was missing in the knockout cell lines”. If not sure include a FH control in the WB or mention that GT103 recognized FH.

We are sure it is CFH, since the band is not present in a genetic knockout of CFH. Furthermore, no other bands in EVs are recognized by GT103. We changed the sentence to be more declarative.

- 200: Binding of GT103 to the exosome-bead conjugates was significantly greater than binding to control antibody > binding of? Changed “to” to “of”.

- 317: “Upon classical pathway activation, immune complexes are formed on the target cell surface”. Immune complexes are known to activate the classical pathway but if this true please, include a reference. We added a reference for the classical pathway (revision line 324).

- “In vitro, the antibody operates predominantly by the classical pathway, initiated by the formation of immune complexes (unpublished data).” According to journal guidelines, unpublished work should not be cited. The data can be included as supplementary material or deposit the data in a publicly available database.

These data have now been added as S3 Fig.

- 347-350: Please, include references We added 3 references for the mechanisms of CFH and CD59 (revision line 395).

- Fig. 3. Why there is no CFH in early-stage EVs? Did the authors detect early-stage EVs? If they do not bind FH do they express less sialic acids or another ligand? The presence of sialic acids would be very easy to check in flow cytometry by for example using fluorescent labeled Maackia Amurensis Lectin I (MAL I). This should be discussed in the manuscript.

If we had found that early stage EVs have no CFH, the sialic acid experiment would be very interesting. However, we couldn’t detect EVs in early stage patients in the first place so we cannot investigate this.

Reviewer #2: This paper by Bushey et al. examines whether EVs from cancer cells express factor H on their surface, and whether an antibody that targets factor h causes the EVs to be lysed. The manuscript is well clearly written and the data clearly presented. The data is all in vitro, which limits the insights into what effect this treatment would have on cancer in vivo. It is strengthened, though, by examination of samples from cancer patients. I have the following questions.

1. The phagocytosis assay studies exosome-bead conjugates. It does not seem that this assay will reflect what happens to free exosomes.

We agree that exosome-bead conjugates are not the same as exosomes. Exosomes were conjugated to beads to facilitate handling during the flow cytometry experiment. (This technique was used previously by Clayton et al. 2003, as cited in the manuscript.) Although exosome-bead conjugates are an artificial construct, this was nevertheless a controlled experiment in which GT103, IgG, or no antibody were added to the exosome-bead conjugates that were presented to macrophages. Significant differences were seen between GT103 and the negative controls (at each of two antibody concentrations) and so we thought this was an interesting finding to present.

2. Figure 2 – the authors find greater factor H on EVs from metastatic cancer cell lines than less metastatic cells. Presumably the factor H came from the parent cells themselves. But since the EVs will be surrounded by plasma FH. But perhaps EVs from the other types of cancer would have factor H on them as soon as they are exposed to serum. The authors could test this, then expose the EVs to serum, wash, and see if any factor H bound. The question seems to be whether the cancer cell produces factor H, or whether there is some other compositional difference of the EVs that makes factor H stick to them.

This is an interesting comment about an issue that we have thought about. Although beyond the scope of this paper, we intend to address this question in future studies.

3. The study found very few EVs in samples from early stage cancer patients. Some studies have reported high numbers of exosomes in normal patient plasma (e.g. Huang et al. BMC Genomics 2013;14:319. ) Is there something different about the isolation process that accounts for this? The authors used a commercial column to purify the EVs, but perhaps some characterization of the EVs as outlined in Théry, et al., 2018; DOI: 10.1080/20013078.2018.1535750 would he useful.

In Fig. 3 and S2 Fig., we found undetectable levels of CFH-containing EVs in 9 of 10 early stage cancer patients as well as 3 normal patients (Two normal patient samples were cut off the original Fig. 3 western blot and were restored for this resubmission. There is an additional normal patient sample in S2 Fig.) In contrast, we found relatively high levels of CFH-containing EVs in 9 of 9 late stage cancer patients. We did not use a commercial column to isolate EVs; we isolated the EVs by ultracentrifugation. This is a standard technique in the field. We characterized the EVs by NTA, also standard in the field. It doesn’t seem likely that the isolation procedure would account for the difference between early and late stage. 

4. In figure 3 could probe the blot for CD63 and see if any exosomes were isolated from the early stage samples. NTA did not show any EVs or exosomes in the early stage samples whereas they were abundant in the late stage samples. Because of this we didn’t think probing the blot for CD63 would reveal anything. 

5. Lysis of the exosomes seems to involve the classical pathway. This suggests that blocking factor H function is less important than binding to an EV surface protein. Factor H has a fairly low affinity for most surfaces, so perhaps a transmembrane protein would be a better target. 

This is all true. We initially thought that blocking CFH function would trigger the alternative pathway. But we found that, although there was some alternative pathway activity, binding of an antibody to CFH as an EV surface protein and triggering the classical pathway accounted for most of the exosome lysis. We were able to show this in spite of the fact that CFH has low affinity for surfaces. In the Discussion, we do consider other types of targets.

6. Are the authors certain that the antibody does not bind to factor H on EVs from other non-cancer cell types? They could test binding to EVs isolated from other cell lines (e.g. macrophages). The authors have evidence that the antibody is specific for factor H on cancer cells, but if it did bind to factor H on healthy cell-EVs it could disrupt physiologic processes.

Yes, as stated above, we looked at GT103 binding to, and lysis of, PBMCs in comparison to a tumor cell line. These results are now reported in S1A and S1B.

---

## [Editor Report · Decision Letter 1]

19 May 2021

Complement Factor H Protects Tumor Cell-Derived Exosomes from Complement-Dependent Lysis and Phagocytosis

PONE-D-21-08454R1

Dear Dr. Patz,

We’re pleased to inform you that your manuscript has been judged scientifically suitable for publication and will be formally accepted for publication once it meets all outstanding technical requirements.

Kind regards,

Marco Trerotola

Academic Editor

PLOS ONE

---

## [Editor Report · Acceptance letter]

21 May 2021

PONE-D-21-08454R1 

Complement factor H protects tumor cell-derived exosomes from complement-dependent lysis and phagocytosis 

Dear Dr. Patz Jr.:

I'm pleased to inform you that your manuscript has been deemed suitable for publication in PLOS ONE. Congratulations! Your manuscript is now with our production department. 

Kind regards, 

on behalf of

Professor Marco Trerotola 

Academic Editor

PLOS ONE